# Towards Extending Dividend Puzzle Debate: What Motivates Distribution of Corporate Earnings in Tanzania?

**Josephat Lotto**

The Institute of Finance Management, Dar Es Salaam 11101, Tanzania; josephat.lotto@ifm.ac.tz

**Abstract:** This paper investigates the determinants of dividend policy in Tanzania. The study employed a panel data of non-financial firms listed on the Dar es Salaam Stock Exchange (DSE) for the period 2008–2017. The paper reports profitability, liquidity, firm size, leverage, firm growth, previous dividend, and GDP as the major determinants of corporate dividend policy. According to the results, leverage, firm growth, and GDP are negatively related to dividend payout ratio while firm size, profitability, liquidity, and lagged dividend are positively related to dividend policy. More specifically, large-sized firms, highly profitable firms, and firms who paid dividend in previous years are more likely to consider paying dividend. However, payment of dividend will all depend on whether the firm is liquid enough to afford that. On the other hand, high-growth and leveraged firms would not probably consider paying dividend, and will, therefore, opt saving money to finance their expansion and honor their debt obligations. Following these results, corporate managers are advised to consider preferences of investors towards developing corporate dividend policy; to strive paying dividend whenever economically viable (as it signals the firm's reputation), and to limit excessive borrowing to protect firms from getting into financial meltdown (although borrowing is considered a control tool for agency-related problems).

**Keywords:** dividend payout; leverage; profitability; growth; liquidity; dividend puzzle

## 1. Introduction

Dividend policy is one of the most debated topics and a fundamental theory of corporate finance, which still reserves its prominent place. More than three decades ago, Black (1976) described dividend policy as a "puzzle", and from there-on several authors engaged in attempting to solve the dividend puzzle. The question of why firms pay dividends from their earnings remains unexplained. This is known as the dividend puzzle in finance literature, as pronounced by Alam Khan and Rashid (2009). Many hypotheses have been drawn to shed some light on this puzzle, but the problem still exists. Allen et al. (2000) confessed that, regardless of several theories put in place to explain their pervasive presence, dividends remain one of the persistent puzzles in corporate finance.

Black (1976) stated that "The harder we look at the dividend picture, the more it seems like a puzzle, with pieces that don't fit together", and confirmed that dividend policy is one of the highly ranked unresolved problems in finance literature, and he further insists the lack of adequate explanation for the observed dividend behavior of the firms. The same has been appreciated by Brealey and Myers (2003, 2005). Since John Lintner (1956) and Miller and Modigliani (1961) came up with this debate, some of the questions, which remain unresolved to-date, include: what are the factors that determine dividend policy? Moreover, is dividend policy determined dependently or independently?

Several theoretical and empirical works on dividend policy (e.g., Lintner 1956; Gordon 1959; Miller and Modigliani 1961) are available, but a consensus has not been reached on the debate. The

good news is that, most of the authors consistently agree on the importance of corporate dividend policy on enhancing corporate value. An attempt to examine the factors that determine dividend policy has attracted massive empirical literature, most of which are from developed countries, and few are from emerging economies; yet, many studies indicate conflicting results. For example, Nuhu Eliasu (2014) and Pandey and Ashvini (2016) show that earnings and liquidity are positively related to the dividend payout ratio, while Zameer et al. (2013) and Almeida et al. (2015) reveal that firms with increased earnings have little dividend payout.

Still, questions on how companies set their earnings distribution and why they pay dividends impose a problem in dividend policy, Baker et al. (2002). Allen et al. (2000), in their own words, concluded that, "Although a number of theories have been put forward in the literature to explain their pervasive presence; dividends remain one of the thorniest puzzles in corporate finance". Therefore, lack of a compromising solution for dividend policy attracts more work on this debatable area. Unlike in developed countries, very few studies on dividend policy have been conducted in emerging economies such as Tanzania. The findings of the developed economies may not be directly applied to developing economies, such as Tanzania, due to differences in regulations, culture environment, and nature of investors, as previously suggested by Chay and Suh (2009). Meanwhile, it is not clear why dividend is paid and what are the factors to be considered before corporate managers decide to pay dividend or not. Various studies from different countries, economies, and business environments have conducted research on dividend policy, but due to the variation in legal frameworks, the tax, and the accounting policies among countries, and across industries, no unified way of setting out dividend policy has yet to be established. In Tanzania, few studies—including work conducted by Zawadi Ally (2015), Ngole (2015), Gwahula and Mnyavanu (2018), and Epaphra and Nyantori (2018)—focused on all firms listed on the Dar es Salaam Stock Exchange (DSE), contrary to this study, which focuses only on non-financial firms only. This study suggests that financial firms are different from non-financial firms regarding their dividend payment regulations payments; moreover, the accounting treatment of some of the accounting variables of these firms do differ, and, therefore, placingall of these firms in one basket may amount to some inconsistencies.

In addition, the previous studies in Tanzania focused only on firm-related factors. In practice, dividend policy is not only dependent on internal factors, but, rather, the complete set of both internal and external factors, such as GDPs, interest rates, exchange rates, and a country's inflation level. This study contributes to the debate by uncovering both internal and external factors that affect corporate dividend policy. This study aims at identifying factors that influence dividend policy of all non-financial firms listed on the Dar es Salaam Stock Exchange (DSE), while focusing on agency theory and signaling hypothesis. The DSE is currently a fast growing ideal frontier market, which presents both foreign and local investors with massive bargain opportunities. It offers foreign investors exposure to the Tanzanian economy, and because many listed firms have expanded beyond Tanzania's borders, it also serves as an entry point to the regional economy. In the short term, foreign investors can capitalize by investing in the weak Tanzanian shilling and seek exit points as it strengthens against the US dollar. The findings of this paper have implications for corporate finance and governance theories, academics, investors, regulators, and policymakers in emerging markets. The rest of the paper is organized as follows: in the next section, the related literature and hypotheses are developed, followed by another section, which describes data and methodology. Furthermore, the empirical results are presented in the fourth section and, finally, the last section brings the paper to an end with a concluding remark.

*Related Literature and Hypothesis Development*

A handful theoretical and empirical studies on dividend policy are available. The issue of the dividend is very crucial in the financial market due to various reasons; first, dividends act as a signal used by the public to reflect the firm's financial stability and growth prospects, Ham et al. (2019). Secondly, dividend policy plays a very essential role in determining the corporate capital structure.

The theoretical position manifested by Lintner (1956) reveals that dividends are paid out of profits; that is why it is impossible for an unprofitable company to pay dividends. Literature suggests several drivers of the corporate profitability, such as good corporate governance practices, (Ahmad 2015), high investor protection, (Boţoc and Pirtea 2014), close corporate monitoring by institutional shareholders, (Chang et al. 2016; Jacob and Lukose 2018), regulatory consideration, (Kasozi and Ngwenya 2015), and minimum agency problem (Jacob and Michaely 2017). From these authors, one may deduce that companies with good corporate governance, sound regulatory support, high investor protection, strong corporate monitoring by institutional shareholders, and minimum agency problems are more profitable as compared to those companies with poor corporate governance practices, lower investor protection, weak corporate control by institutional shareholders, and having agency problems. According to Lintner (1956), a firm's net earnings are an important factor influencing dividend payments. On the other hand, highly profitable firms will have greater ability to pay dividends. The view of some proponents of dividend policy is that market share price of a company responds to its declared changes in dividend policy, and this response is resulted from information content in dividend changes. This is supported by Myers and Majluf (1984) who claims that within the pecking order preferred by managers for internal financing, dividend policy is affected by profitability.

Other authors who have tested this contention have confirmed that profitability affects dividend policy; they include Pruitt and Gitman (1991), who report that current and past year profits are important factors influencing dividend payments; Baker and Powell (2000), confirm that anticipated level of future earnings is the major determinant of dividend policy; Darling (1957) concludes that corporate dividend policy usually changes with the change in its past profits, current profits, and expected future profits. Similarly, Al-Ajmi and Hussain (2011) realized that, present and previous year profits do influence payment of dividend. Furthermore, several earlier empirical works from developed economies, such as Jensen et al. (1992), and Fama and French (2001), have reported a positive relationship between profitability and dividend payouts, and empirical evidence from emerging countries, such as Pandey and Ashvini (2016). Aivazian et al. (2003) also support the direct association between corporate profitability and dividend policy. However, in studies, such as one conducted by Ahmad et al. (2018), profitability doesnot directly link with corporate dividend payout. Therefore, following the preceding arguments, one may formulate the following hypothesis:

**Hypothesis 1 (H1).** *Profitability has a positive and significant impact on dividend payout policy.*

The earlier studies on dividend policy, such as Fama and French (2001), recognize the influence of corporate growth on the choice of dividend policy. According to their view, the dividend policy is dependent on whether the firm has any available investment opportunity, and the relationship between the internal rate of return of the firm and its cost of capital. According to Fama and French (2001), argument growing firms, whose internal rate of return is greater than its cost of capital, would have sufficient profitable investment opportunities and, therefore, payment of dividend will not be considered as the priority because doing so would not be consistent with the value-maximization principle. However, in case of declining firms, where internal rate of return is less than cost of capital, the firm will be maximizing value of share by distributing a hundred per cent of earnings as dividends to their shareholders.

Mueller (1972) proposed a formal theory that a firm has a relatively well-defined life cycle, which is fundamental to the firm life cycle theory of dividends. The theory explains that, as firms pass through the various stages in their lives, they tend to alter the dividend policy depending on the financial needs of each stage. Implied in this theory is the fact that firms that are in their growth stages are less likely to pay more dividends as compared to firms that are at their maturity stages. Old firms, therefore, because they do not have a lot of growth opportunities to fund, are expected to pay more dividends.

According to Fama and French (2001), a growing company is presented with the number of promising opportunities, which require capital to finance, and retained earnings is one of corporate's

cheapest sources of project financing. An empirical study of India by Kumar and Sujit (2016) confirms growth opportunity as a major determinant of dividend payout. It therefore follows that high growth companies pay low dividends, or adopt no dividend policy, and low growth firms do pay more dividends. Based on these arguments, the following assertion is proposed:

**Hypothesis 2 (H2).** *Firm's growth has an inverse and significant impact on dividend payout policy.*

Capital structure has long been considered as one of the factors with a strong impact on dividend policy. According to Pal and Goyal (2007), the demand for external finance by the company usually comes into place because of the financial limitation from the internal sources of the firm. Rozeff (1982) also recognizes dividend policy as a determinant of external financing cost, financial constrains resulted from the financial leverage, and the agency cost of outside ownership. According to Rozeff (1982), a highly leveraged firm pays low dividend to their shareholder due to cash flow obligations to their financiers. Similar findings supporting an inverse relationship between leverage and dividend yield is reported by Ahmad et al. (2018). When a firm secures finance through leverage, it accumulates a fixed financial obligation, including payments of interest and the principal. This means that the firm needs to sustain enough cash to pay for those obligations, which will lower the amount of un-distributable profit. That is why a high level of financial leverage results in low dividends payments, Al-Malkawi (2007). The higher the internal flows are given the investment requirements; the lesser will be the demand for borrowing, and vice-versa. Thus, the higher the dividend, the higher the borrowing demand will be. Baker et al. (2002) also pointed out that firms with less external financing in their capital structure experience smaller dividend payout ratios, and that firms with higher levels of debt need higher liquidity to allow payoffs on potential implicit claims. To avoid such costs, firms normally rely on equity as their alternative source of finance. Based on the preceding discussions, the following proposition can be put forward:

**Hypothesis 3 (H3).** *Leverage has a negative and significant impact on dividend payout policy.*

Another vital factor thatinfluences corporate earning distribution is liquidity. It is very imperative to compare a firm's liquidity position with its payment of dividend. Rationally, a firm will choose to pay dividend as long as its cash position is not questionable, and that availability of the profit itself doesnot merit payment of dividend. Liu and Hu (2005) proposed that firms have residual cash when its dividend is less than the free cash flow, otherwise they need additional financing to meet the cash dividend requirement. A weak corporate liquidity position implies less sufficient cash dividend due to cash shortage.

Alli et al. (1993) argue that dividend payments depend more on cash flow (which reflects the company's ability to pay dividends) than on current earnings, which are less heavily influenced by accounting practices. While Amidu and Abor (2006) found a direct and significant relationship between cash flow and dividend payout ratios, Anil and Kapoor (2008), on the other hand, consider cash flow an important determinant of dividend payout ratio. Following these arguments, one can set out the following proposition:

**Hypothesis 4 (H4).** *Liquidity has a positive and significant impact on dividend payout policy.*

Literature, such as Al-Najjar and Hussainey (2009) recognizes firm size as a crucial factor that influences corporate dividend policy. Others, such as Ho (2003), argue that large-sized companies have more ability to pay dividends, rather than smaller-sized ones. This is consistent with Aivazian et al. (2003) who compounded that the larger firms have easy access to the financial market for possible project financing, compared to smaller firms; therefore, payment of dividend may not be a serious constraint because they are not faced with financial limits when it comes to financial potential profitable investments. This view is in line with Adedeji (1998), who appreciates the ability of large firms to

secure easily and cheaply external financial sources for funding new projects. These arguments result into the following propositions:

**Hypothesis 5 (H5).** *Firm size has a positive and significant impact on dividend payout policy.*

Over time, as goods and services become more expensive, the value of money will subsequently fall, and the purchasing power of people will also come down. This is the situation in an inflationary environment. Inflation can be defined as the persistent rise in aggregate level of prices of goods and services in an economy. According to Adrangi et al. (1999), consistent price rise wears out the purchasing power of money and other financial assets with fixed values, creating serious economic distortions and uncertainty, and they point out that some portion of inflation rate will be anticipated by economic agents and capital markets. However, the unanticipated portion of inflation may surprise equity markets and affect returns. McGuigan and Kretlow (2009) argue that in an inflationary environment, funds generated by depreciation often are not sufficient to replace a firm's assets as they become obsolete. Under these circumstances, a firm may be forced to retain a higher percentage of earnings to maintain the earning power of its asset base. Following this discussion, one may put forward the following proposition:

**Hypothesis 6 (H6).** *Inflation has a negative relationship with the dividend payout ratio.*

Classical economic fluctuation concerns the absolute volatility of economic output; e.g., gross domestic product (GDP) declines during an economic downturn. Growth economic fluctuation, however, is based on changes in the economic growth rate. The GDP growth rate represents the market value of all the goods and services produced within the boundary of a country in a specified period of time. When the real economic activity of the economy increases, it leads to increase the corporate earning of the different companies, which ultimately leads to increase the dividend payout ratio (Ghafoor et al. 2014). Based on these arguments, one may come up with the following hypothesis:

**Hypothesis 7 (H7).** *Gross Domestic Product (GDP) has a positive relationship with the dividend payout ratio.*

## 2. Methodology

### 2.1. Data and Sampling Strategy

This paper employed secondary data, which have been collected from the audited annual reports of listed firms in the Dar es Salaam Stock Exchange (DSE) for a period between 2008 and 2017. The annual reports have been sourced from the website of the listed firms, DSE publications, and database of African listed firms' annual reports. A company was selected as long as it had complete accounts from 2008. Most companies lacked some information required prior to 2008, so the period before 2007 would not be useful for this purpose. The criteria by which the firm was included in the sample are: (i) the firm must have had available data for all years, from2008 to 2017; (ii) The firm must have been listed continuously on DSE before the aforementioned period of time; (iii) The firm must have paid dividends in all years across the period 2008–2017

Moreover, the time span between 2008 and 2017 is sufficient, because within this time period, about two business cycles were completed, as proposed by Rafique (2012), who suggests that one business cycle is complete in 5–7 years.

Out of the 27 firmslisted in the Dar es Salaam Stock Exchange, 13 financial firms (banks and other financial institutions) were excluded from the sample because their regulations regarding dividend payments are different from other firms, such as manufacturing and industrial firms, and also because accounting treatment of some of the accounting variables are different. The exclusion of the financial firms is also proposed by von Eije and Megginson (2008). Out of the remaining sample of 14 firms,

two of them were removed from the sample, as either they did not have complete information or they were not listed in the exchange market continuously for the period of the study. In general, only 11 non-financial firms continuously listed for the period of the study, and were making profit, qualified to form a composition of the sample of this study. Most of these selected firms are foreign firms listed in DSE and they have also existence in other emerging economies. Based on this fact, one may believe that factors influencing these firms to pay dividends are similar across emerging economies because most stock markets in emerging economies are still in their infant stage, and, therefore, firms operating in such markets may likely have similar behaviors, as far as dividend decision is concerned. This may be due to the fact that growing capital markets may not be liquid enough to provide capital avenues to the firms, so, most likely, internal financing (retention) becomes the only available alternative source of capital.

### 2.2. Descriptive Statistics

Table 1 also shows an average dividend payout ratio of about 19% for all firms sampled in this paper. It also shows, in Table 1, that the maximum and minimum dividend payout ratios for the samples are0.86 and 0.04, respectively. This shows that all sampled firms in the study are paying dividends at the minimum rate of 4%. Furthermore, Table 1 shows that firm size has a minimum of 0.11 and a maximum of about 19.52, with an average (mean) of about 0.44. It also shows that firm profitability, measured as return on equity, has a mean value of about 45%, with maximum of 98% and minimum of 6%, as shown in descriptive statistics Table 1. Likewise, the descriptive statistic Table 1 shows that the minimum and maximum growth rate of sampled firms are 1.2% and 77%, respectively, while average growth rate stands at 17%. The table further depicts that firms' average liquidity is about 66%, with a minimum of 13% and maximum of 98%

**Table 1.** Descriptive Statistics.

|  | Obs. | Mean | Std. Dev. | Minimum | Maximum |
|---|---|---|---|---|---|
| **DPO** | 121 | 0.1921 | 0.1538 | 0.0410 | 0.8635 |
| **GROW** | 121 | 0.1560 | 0.9754 | 0.0123 | 0.7659 |
| **LIQ** | 121 | 0.6576 | 2.4764 | 0.1327 | 0.978 |
| **LEV** | 121 | 0.4922 | 0.1386 | 0.0045 | 0.87260 |
| **PROF** | 121 | 0.4484 | 0.2487 | 0.0570 | 0.9763 |
| **SIZE** | 121 | 4.4041 | 6.2759 | 0.1123 | 19.5245 |

Note: DPO = Dividend payout; PROF = Profitability; LIQ = Liquidity; GR = Growth; SIZE = Firm Size; LEV = Leverage.

### 2.3. Analytical Design

This paper used a panel data analytical design, which involves the pooling of observations on a cross-section of firms listed in DSE over a period of ten years. To check the significance of relationship between dividend payout and explanatory variables (profit, liquidity, growth, leverage, firm size, GDP per capita, inflation), the study employed the random effect regression analysis; to examine the likeliness of the proposed hypotheses, and the significance of the individual explanatory variable, t-test was used. Before running multiple regressions, panel data were subjected to regression diagnostic tests, such as multicollinearity and heteroscedasticity.

### 2.4. Regression Diagnostic Tests

Before running regression, we performed regression diagnostic tests, such as multicollinearity and heteroscedasticity. Multicollinearity, according to Jurczyk Omáš (2012), is a condition where the explanatory variables are virtually linear dependent. In Table 2 we can observe that the highest correlation among all the variables is +0.57, which is the correlation between inflation and GDP.

However, an absolute value larger than 0.8 is preferred to be enough to cause multicollinearity, as recommended by Studenmund (2011). Considering that +0.57 is far from 0.8, we conclude that there is no problem of multicollinearity among our variables.

**Table 2.** Correlation Matrix for the dividend payout (DPO) regression.

|  | DPO | PROF | LEV | SIZE | GROW | LDPO | LnGDP | LIQ |
|---|---|---|---|---|---|---|---|---|
| **DPO** | 1.000 | | | | | | | |
| **PROF** | 0.179 | 1.000 | | | | | | |
| **LEV** | −0.520 | −0.188 | 1.000 | | | | | |
| **SIZE** | −0.184 | −0.105 | 0.121 | 1.000 | | | | |
| **GROW** | 0.014 | 0.047 | 0.048 | 0.129 | 1.00 | | | |
| **LDPO** | 0.217 | 0.116 | −0.343 | −0.192 | −0.153 | 1.00 | | |
| **LnGDP** | 0.1491 | −0.084 | 0.112 | 0.123 | −0.009 | 0.228 | 1.00 | |
| **PI** | −0.126 | 0.089 | −0.118 | −0.069 | −0.009 | −0.097 | −0.572 | |
| **LIQ** | 0.014 | 0.047 | 0.048 | 0.129 | 0.075 | 0.067 | 0.05 | 1.00 |

Alternatively, to test whether there is a potential multicollinearity, we employ the Variance Inflation Factor (VIF), where a value of VIF exceeding 10 implies a potential problem, as advocated by Belsley et al. (1980). The VIF test shows that there is no problem related to multicollinearity because the VIF value is just 3.35, as shown in Table 3.

**Table 3.** A test of Multicollinearity using Variance Inflation Factor (VIF) Test.

|  | VIF | 1/VIF |
|---|---|---|
| **LIQ** | 5.06 | 0.1976 |
| **GROW** | 3.53 | 0.2833 |
| **LEV** | 3.11 | 0.3215 |
| **PROF** | 2.25 | 0.4444 |
| **SIZE** | 3.15 | 0.3175 |
| **PI** | 3.12 | 4.3205 |
| **LNGDP** | 3.25 | 0.3077 |
| **Mean VIF** | 3.35 | |

After the test for multicollinearity, we also performed a Wald test for heteroscedasticity. The concern of heteroscedasticity is the homogeneity of variance of the residuals. This is one of the conditions to be met before normal regression is run. The results of Wald test are presented in Table 4. The results demonstrate a Chi value that is greater than the critical value, meaning that we could reject the hypothesis for homoscedasticity.

**Table 4.** Wald test for heteroscedasticity.

| Modified Wald test for groupwise heteroscedasticity in random effect regression model<br>HO: *sigma (i) ^2 =sigma ^2 for all i*<br>Variables: *fitted values of dividend payout ratio* | |
|---|---|
| Chi2(8) | **258.25** |
| Prob >chi2 | **0.000** |

According to Huber (1980), the homoscedasticity assumption is needed to show the efficiency of OLS. From the test conducted, we reject Nul hypothesis and accept Alternative hypothesis at Probability>chi2=0.000, and conclude that there is heteroscedasticity. Thus, the usual regression t-statistics and confidence intervals are no longer valid for inference problem. Using regression estimator without adjustment will render estimations biased. To solve this problem, the regression estimators are improved by finding heteroscedasticity-robust estimators of the variances using a fixed effect robust regression method.

*2.5. Model Specification*

In specifying the empirical model, the independent variables may explain much of what is different about an observation, a firm, or a year, but there is probably some unmodeled heterogeneity. According to Lotto (2018), usually, the heterogeneity, which is left unmodeled, goes into the error term. The true problem occurs when some firms (or, less commonly, time periods) share some unmodeled heterogeneity. In this case, we would like to be able to explain everything that makes each firm different, but usually this is unmanageable, so something must be done to eliminate this shared and, thus, systematic heterogeneity from the error term. Since this study uses panel data, to solve the potential problem of heterogeneity, either a fixed effect or random effect regression model should be employed. To decide between fixed or random effects, a Hausman test—where the null hypothesis is that the preferred model is random affects vs. the alternative fixed effects, Greene (1980)—is used. To do this, the Hausman test was conducted. The Hausman test shows whether the unique errors are correlated with the regressors; the null hypothesis is that they are not correlated. If the probability of chi-squared in the Hausman test output is less than 0.05, fixed effects is preferred, otherwise random effect is preferable. When this test was run, the Chi-squared was found to be 0.0876, which is greater than 0.05; hence, the study chose to apply the random effect regression model. The random effects model considers the differences between individual firm effects. The rationale behind the random effects model is that, unlike the fixed effects model, the variation across firms is assumed to be random and uncorrelated with the predictor or independent variables included in the model. The result of the Hausman test is presented in Table 5

**Table 5.** Hausman Test.

| | Coefficients | | (b-B) | Sqrt(diag(v_b-v_B)) S. E |
|---|---|---|---|---|
| | *(b)* *fe* | *(B)* *re* | | |
| **PROF_d1** | −0.001 | 0.000 | −0.001 | . |
| **LEV** | −0.200 | −0.351 | 0.151 | 0.165 |
| **SIZE** | −0.593 | −0.01 | −0.583 | 0.185 |
| **GROW** | 0.060 | 0.037 | 0.022 | . |
| **LDPO** | −0.902 | −0.688 | −0.224 | . |
| **LnGDP** | 1.016 | 0.365 | 0.651 | 0.203 |
| **LIQ** | 0.050 | 0.047 | 0.052 | 0.034 |
| **PI** | −0.000 | 0.000 | -0.000 | . |

B = Consistent under $H_0$ and $H_a$; obtained from xtreg. B = inconsistent under $H_a$; Efficient under $H_o$; obtained from xtreg. Test: Ho: Difference in coefficient not systematic. Chi (8) = 12.42. Prob>chi2= 0.0876. v_b-v_B is not positive definite.

The empirical model takes the following form:

$$Y_{it} = \alpha + \beta\ X_{it} + \mu_{it}$$

where $y_{it}$ is the dependent variable, $\alpha$ is the intercept term, $\beta$ is a kx1 vector of parameters to be estimated on the explanatory variables, and x is a 1 x k vector of observations on the explanatory variables. The variable description is provided in Table 6 below.

**Table 6.** Variables, Description and Expected relationships.

| Variable | | Description | Expected Sign |
|---|---|---|---|
| Dependent Variable | Dividend Payout ratio | Total dividend/Net Income | |
| | Profitability | Net income/Total equity | Positive |
| | Leverage | Debt/Assets | Negative |
| | Lagged dividend | Previous year dividend/share equity | Positive |
| Independent Variables | GDP per capita | LnGDPpc | Positive |
| | Firm size | Natural log of total assets | Positive |
| | Liquidity | Current Assets/Current Liabilities | Positive |
| | Growth | (Current sales–previous sales)/previous sales | Negative |
| | Inflation | Consumer price index | Negative |

The following multiple regression model is specified;
DPO = $f$ ((PROF, GR, SIZE, LEV, LIQ, LDPO, INF, GDP))

$$DPO = \beta_0 + \beta_1 PROF + \beta_2 LEV + \beta_3 LDPO + \beta_4 SIZE + \beta_5 GR + \beta_6 LIQ + \beta_7 INF + \beta_8 GDP + \mu$$

where;

DPO = Dividend payout
PROF = Profitability
LIQ = Liquidity
GR = Growth
SIZE = Firm Size
LEV = Leverage
LDPO = Lagged dividend Pay out
INF = Inflation
GDP = Gross domestic Product per capita
$\mu$ = error term

## 3. Empirical Evidence

This section discusses the findings presented in Table 7 below. Table 7 shows the regression results of pooled ordinary least squares, random-effects, and fixed-effects models on determinants of dividend policy. As previously pointed out, the Hausman test specifies that the random-effects model fits more for this study. Similarly, as shown in Table 7, the random-effects model is the best model to explain the factors affecting dividend policy, because it records the highest adjusted$R^2$ value of 71%. This shows that the nine factors examined in this paper explain about 71% of the determinants of dividend policy. Of these factors, both leverage and firm growth are negatively statistically significant at 5% significant level, while firm size, profitability, liquidity, and previous year dividend all have positive statistically significant relationships with the dividend payout ratio at 5%, 1%, and 10% significant level, respectively.

**Table 7.** A regression results table.

| Regressor | Pooled Regression Model | | Random Effect (Robust Model) | | Fixed Effect (Robust Model) | |
|---|---|---|---|---|---|---|
| Regressor | Coeff. | *t*-stat | Coeff. | *t*-stat | Coeff. | *t*-stat |
| PROF | 0.133 | 3.45 *** | 0.057 | 4.23 *** | 0.393 | 2.87 ** |
| LEV | 0.192 | 1.57 | −0.092 | −2.12 ** | −0.152 | |
| SIZE | −0.051 | −1.67 * | 0.035 | 2.12 ** | −0.041 | −1.75 * |
| GROWTH | 0.024 | 1.12 | 0.039 | 2.45 ** | 0.094 | 1.94 * |
| LDPO | 0.033 | 2.64 ** | 0.043 | 1.97 * | 0.018 | 2.98 ** |
| GDP | −0.162 | −1.23 | −0.364 | −1.98 * | −0.179 | −2.13 ** |
| LIQ | 0.014 | 1.23 | 0.015 | 3.45 *** | 0.047 | 2.94 ** |
| INF | 0.181 | 1.17 | 0.172 | 1.23 | 0.276 | 1.45 |
| Constant | 0.171 | 0.12 | 0.148 | 2.12 ** | 0.239 | 3.67 *** |
| Adj.R2 | 0.67 | | 0.71 | | 0.63 | |
| F-stats | 145.34 (0.000) | | 58.50 (0.000) | | 187.98 (0.000) | |
| Durbin–Watson | 2.34 | | 4.25 | | 1.87 | |

**Note:** *, ** and *** indicate significant at 10%, 5%, and 1% respectively.

The statistically significant positive relationship shown in Table 5, between profitability and dividend payout, suggests that highly profitable firms will have greater ability to pay dividends. This finding is consistent with the signaling theory of dividends, according to which firms that are more profitable pay handsome dividends, reflecting to the market their better financial performance.

On asimilar vein, a positive relationship, which liquidity holds with dividend payout was expected, because rationally, a firm chooses to pay dividend as long as its cash position has no problem, and that availability of the profit itself does not merit payment of dividend. Liu and Hu (2005) proposed that firms have residual cash when dividend is less than the free cash flow, and that if cash dividend is great than the free cash flow, then that firm needs additional financing to meet the cash dividend requirement. Alli et al. (1993) argues that dividend payments depend more on cash flow, which reflect the company's ability to pay dividends, than on current earnings, which are less heavily influenced by accounting practices. While Amidu and Abor (2006) found a direct and significant relationship between cash flow and dividend payout ratios, Anil and Kapoor (2008), on the other hand, consider cash flow an important determinant of dividend payout ratio.

On the other hand, the paper finds a negative effect of growth (investment) opportunities on corporate earnings distribution decision aligning with the pecking order and transactions cost theories. The message derived from these theories are that high-growth firms require more money to finance their expansion; therefore, they are more likely to save internally generated earnings for financing investment projects rather than paying dividends, because it is more costly to consider external financing options at an expense of paying dividend. In fact, all the firms that experience above-average growth rates are expected to have low dividend payout ratios since, in line with the residual theory of dividends, a greater number of profitable investment opportunities should result (other things being equal) in a greater need for earnings retention. Al-Malkawi (2007), Juma'h and Pacheco (2008), and Foroghi et al. (2011)reported consistent findings.

Likewise, the paper shows that debt level negatively influences dividend policy. This negative relationship implies that the use of debt into corporate capital structure and payment of dividends are considered as substitute tools in controlling agency-related problems. This may as well indicate that highly leveraged firms tend to preserve internal funds to honor their debt obligations and lower external financing costs instead of paying the cash to shareholders as dividend, in line with transaction cost theory. Since the firm's focal priority is to debt holders, the amount of money that would be distributed to shareholders as dividends will depend on the balance available after honoring debt

obligations. This finding is in line with Al-Malkawi (2007), Kowalewski et al. (2007), Ramli (2010), and Al-Shubiri (2011a, 2011b).

Regarding the firm size, the results show that large firms pay higher dividends. This finding is in line with the Fama and French (2001), Holder et al. (1998), and Jensen et al. (1992). It should also be clear that large firms have better access and easier, cheaper ways of raising funds compared to small firms; therefore, other things remain the same, large firms are more likely to afford paying higher dividends to shareholder.

The empirical analysis of this paper also considered the dividend paid during the previous year as the important determinant of current corporate dividend payments. The results presented in Table 7 show that the previous year's dividend payout has a positive and significant relationship with the current dividend payout. This relationship is consistent with the dividend-smoothing hypothesis pioneered by Lintner (1956) and implies that companies increase their dividend payout ratios, referring to previous dividends as a benchmark, and are reluctant to reduce them when they forecast a sustainable future cash flow growth.

Apart from the internal factors, there are also external factors that affect the earning distribution of the firm—among which are inflation and GDP. Table 5 shows a negative significant relationship between dividend payout and GDP, indicating that the higher the GDP per capita, the lower the dividend payout. This implies that in a country where GDP is high, shareholders are less likely to consider or expect dividend payments. During high GDP levels, the economic environment is potentially conducive for investing; therefore, reinvesting the corporate profit is relatively a wise decision than distributing it back to owners as dividend.

Table 7 also shows insignificant positive relationship between inflation and dividend payout. This result is contrary to the expectation that there is a negative relationship between inflation and payout policy. Overall, the paper reveals that factors affecting the corporate dividend decisions of firms listed in the Dar es Salaam Stock Exchange do follow similar patterns to those in more developed economies.

## 4. Concluding Remarks

This paper mainly aimed to examine the determinants of dividend payout policies of selected listed firms in the Dar Es Salaam Stock Exchange using a data set covering the period between 2008 and 2017. According to the empirical results, both leverage and firm growth have a statistically significant negative relationship with corporate dividend payout, while firm size, profitability, and previous year dividend all have positive statistically significant relationships with dividend payout ratio.

The positive relationship between profitability and dividend payout suggests that highly profitable firms will have greater ability to pay dividends—this is generally consistent with a theoretical position manifested by Lintner (1956). Similarly, a positive relationship between a firm's growth opportunity and dividend payout can be explained with the help of signaling theory. In fact, since these firms have positive expectations about the future, they may increase their dividend payments, and exhibit indicative information to their shareholders about their optimistic expectations. Likewise, the results may be due to the possibility that firms with high investment opportunities may have easy access to other external financing options, and do not highly depend on internal financing for future investment.

Regarding the firm size, the results show that large firms pay higher dividends. The paper concludes that large firms have better access and easier cheaper ways of raising funds compared to small firms; therefore, other things remain the same, large firms are more likely to afford paying higher dividends to shareholders.

The paper also reveals a negative relationship between leverage and corporate dividend payout, which means that firms with heavy debt in their capital structures tend to consider servicing their debt obligations as the top priority over dividend payments. Since the firm's focal priority is to paying debt, the amount of money that would be distributed to shareholders as dividends would depend on the balance available after honoring debt obligations.

The empirical analysis of this paper also shows that the previous year's dividend payout has a positive and significant relationship with the current dividend. This relationship is consistent with dividend smoothing hypothesis, which implies that companies increase their dividend payout ratios based on the previous dividends paid to their shareholders, and they become reluctant to reduce the level of dividends previously paid, especially when they have forecasted future cash-flow growth.

While the paper has earmarked the factors that affect corporate managers' decisions on dividend policy, corporate managers are advised to observe the following; First, to consider preferences of investors toward developing corporate dividend policy, because they are the core beneficiaries of the policy. Second, to strive paying dividends whenever economically viable because it signals the firm's positive reputation—and most shareholders believe so much in the bird in-hand theory for dividends. Third, although borrowing is considered a control tool for agency-related problems, managers are advised to limit excessive borrowing, as this may put the firm in financial meltdown.

**Funding:** This research received no external funding.

**Conflicts of Interest:** The author declares no conflict of interest.

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
