# Peer review of "Towards Extending Dividend Puzzle Debate: What Motivates Distribution of Corporate Earnings in Tanzania?"

_ijfs, doi:10.3390/ijfs8010018_

Round 1
Reviewer 1 Report
In this manuscript, author(s) present an empirical investigation that seeks to explore the determinants of dividend policy of listed companies in Tanzania. The manuscript deals with an interesting issue, in one of the most under-studied continent, which potentially could provide interesting implications for both academics and managers. But the present version of this paper does not live up to the expectations it creates and should be improved.
In fact, there are some concerns related to the study as listed below:
- The problem definition and paper's positioning could be improved. The author(s) should provide a better identification of the field in which the present paper anticipates contributing to. In other words, the author(s) should clearly explain the gap in the literature and why it is important to study this small stock exchange.
- Chapter “Related Literature and Hypothesis Development” is not adequate, with some of most recent literature in the field missing. Author(s) should improve information regarding the identification of the main relationships of their theoretical framework
- Suggested additional references:
- Ahmad, G. N. (2015). Does corporate governance affect dividend policy: Evidence from ASEAN emerging market. Risk Governance & Control: Financial Markets & Institutions
- Ahmad, N. G., Barros, V., & Sarmento, J. M. (2018). The determinants of dividend policy in Euronext 100. Corporate Ownership & Control
- Botoc, C., & Pirtea, M. (2014). Dividend payout-policy drivers: Evidence from emerging countries. Emerging Markets Finance and Trade
- Chang, K., Kang, E., & Li, Y. (2016). Effect of institutional ownership on dividends: An agency-theory-based analysis. Journal of Business Research
- Ham, C. G., Kaplan, Z., & Leary, M. T. (2019). Do dividends convey information about future earnings?. Journal of Financial Economics
- Hanlon, M., & Hoopes, J. L. (2014). What do firms do when dividend tax rates change? An examination of alternative payout responses. Journal of Financial Economics
- Jacob, C., & Lukose, J. (2018). Institutional ownership and dividend payout in emerging markets: Evidence from India. Journal of Emerging Market Finance
- Jacob, M., & Michaely, R. (2017). Taxation and dividend policy: The muting effect of agency issues and shareholder conflicts. The Review of Financial Studies
- Kasozi, J., & Ngwenya, A. (2015). Determinants of corporate dividend payment policies: A case of the banking industry in South Africa. Journal of Governance and Regulation
- Kumar, B. R., & Sujit, K. S. (2016). Determinants of dividend policy in GCC firms: An application of partial least square method. Corporate Ownership & Control
- There is very little information about the data collected and the stock exchange in Tanzania. In fact one of weaknesses of this manuscript is the presentation of the sample, with no demographic information about the sample (dimension by sales or employees, capitalization, deb ratio etc.)
These comments and concerns are written to help you to improve the journal quality of your manuscript.
Author Response
Response to Reviewer 1 Comments
Point 1: The problem definition and paper's positioning could be improved. The author(s) should provide a better identification of the field in which the present paper anticipates contributing to. In other words, the author(s) should clearly explain the gap in the literature and why it is important to study this small stock exchange
Response 1: The problem definition and paper's positioning is improved by explaining the gap in the literature and providing the importance of studying DSE. The attached revised version of the paper shows the improvement in LINE 56-87
Point 2: Chapter “Related Literature and Hypothesis Development” is not adequate, with some of most recent literature in the field missing. Author(s) should improve information regarding the identification of the main relationships of their theoretical framework
Response 2: Chapter on Related Literature and Hypothesis development is improved using the proposed literatures by the reviewer. The literature is used to improve several paragraphs of the sections as observed in the attached revised documents in;
- LINE 89-93
- LINE 96-105
- LINE 125-126
- LINE 148-149
- LINE 159-16
Point 3 Suggested additional references:
- Ahmad, G. N. (2015). Does corporate governance affect dividend policy: Evidence from ASEAN emerging market. Risk Governance & Control: Financial Markets & Institutions
- Ahmad, N. G., Barros, V., & Sarmento, J. M. (2018). The determinants of dividend policy in Euronext 100. Corporate Ownership & Control
- Botoc, C., & Pirtea, M. (2014). Dividend payout-policy drivers: Evidence from emerging countries. Emerging Markets Finance and Trade
- Chang, K., Kang, E., & Li, Y. (2016). Effect of institutional ownership on dividends: An agency-theory-based analysis. Journal of Business Research
- Ham, C. G., Kaplan, Z., & Leary, M. T. (2019). Do dividends convey information about future earnings?. Journal of Financial Economics
- Hanlon, M., & Hoopes, J. L. (2014). What do firms do when dividend tax rates change? An examination of alternative payout responses. Journal of Financial Economics
- Jacob, C., & Lukose, J. (2018). Institutional ownership and dividend payout in emerging markets: Evidence from India. Journal of Emerging Market Finance
- Jacob, M., & Michaely, R. (2017). Taxation and dividend policy: The muting effect of agency issues and shareholder conflicts. The Review of Financial Studies
- Kasozi, J., & Ngwenya, A. (2015). Determinants of corporate dividend payment policies: A case of the banking industry in South Africa. Journal of Governance and Regulation
- Kumar, B. R., & Sujit, K. S. (2016). Determinants of dividend policy in GCC firms: An application of partial least square method. Corporate Ownership & Control
Response 3: The literature proposed and used in improving the section on related literature and hypothesis development is included in the reference list as highlighted in yellow
Point 3: There is very little information about the data collected and the stock exchange in Tanzania. In fact one of weaknesses of this manuscript is the presentation of the sample, with no demographic information about the sample
Response 3: The sample presentation is now improved as the demographic information in the form of descriptive statistics is provided in LINE 226-265 of the attached revised version of the paper
Reviewer 2 Report
The main purpose of this paper is to investigate the determinants of dividend policy in Tanzania. The study uses a panel data of non-financial firms listed on DSE from 2008 to 2017. This paper contributes to the debate by uncovering both internal and external factors which affect corporate dividend policy. The author reports profitability, liquidity, firm size, leverage, firm growth, previous dividend, GDP and inflation as the major determinants of corporate dividend policy. According to the empirical results, both leverage and firm growth have a significant negatively relationship with corporate dividend payout while firm size, profitability, and previous year dividend all have positive significant relationship with dividend payout ratio.
This is an interesting paper which makes contributions towards related literature. I list my comments as follows.
1. The author should provide a powerful motivation in Introduction section. Needless to say that the fact that nobody has studies this issue before is insufficient to be the sole motivation of such a study. In fact, all hypotheses in this study have been examined by previous papers even empirical literature from developing countries. The author needs to answer the question “why Tanzania”? What is the difference in Tanzania from other emerging markets? It makes this research look like an exercise.
2. Empirical results of the study are based on only 11 firms. You should explain how those samples can represent emerging economies well. According to your selection criteria, I guess most firms are large-sized and old companies. I suggest you to provide the descriptive statistics for all variables.
3. Since Zameer et al. (2013) and Almeida et al. (2014) reveal that firms with increased earnings have little dividend payout, you should explain why you have conflicting results in this research.
4. In Line 217, the author mentions that the coefficient of correlation between inflation and GDP is +0.57. However, the correlation is -0.57 in Table 1. Although +0.57 is far from 0.8, it is also high degree of correlation. Your regression model might have the problem of multicollinearity. The author should use VIF to check the model.
5. In Line 99, I can’t find “Pandey (2001)” in your References as well as “Adrangi, Chatrath and Sanvicente (2000)” in Line 171 and “McGuigan, Kretlow and Moyer (2009)” in Line 175.
I hope that my comments will help the authors to improve the quality of the paper.
Author Response
Response to Reviewer 2 Comments
Point 1: The author should provide a powerful motivation in Introduction section. Needless to say that the fact that nobody has studies this issue before is insufficient to be the sole motivation of such a study. In fact, all hypotheses in this study have been examined by previous papers even empirical literature from developing countries. The author needs to answer the question “why Tanzania”? What is the difference in Tanzania from other emerging markets? It makes this research look like an exercise.
Response 1: The motivation of the paper is improved by explaining the gap in the literature and providing the importance of studying Tanzania. The attached revised version of the paper shows the improvement in LINE 56-87
Point 2: Empirical results of the study are based on only 11 firms. You should explain how those samples can represent emerging economies well. According to your selection criteria, I guess most firms are large-sized and old companies. I suggest you to provide the descriptive statistics for all variables
Response 2: The explanation of whether sample represent emerging economies is presented in LINE 226-250 of the attached improved version of the paper while the descriptive statistics is provided in LINE 253-265
Point 3: Since Zameer et al. (2013) and Almeida et al. (2014) reveal that firms with increased earnings have little dividend payout, you should explain why you have conflicting results in this research.
Response 3: The argument on the conflicting results in this research are provided in LINE 362-368 of the improved version of the attached paper
Point 4: In Line 217, the author mentions that the coefficient of correlation between inflation and GDP is +0.57. However, the correlation is -0.57 in Table 1. Although +0.57 is far from 0.8, it is also high degree of correlation. Your regression model might have the problem of multicollinearity. The author should use VIF to check the model.
Response 4: The regression model is confirmed to have no multicollinearity problems as suggested by the VIF test followed by explanation presented in LINE 285-290 of the attached revised version of the paper
Point 5: In Line 99, I can’t find “Pandey (2001)” in your References as well as “Adrangi, Chatrath and Sanvicente (2000)” in Line 171 and “McGuigan, Kretlow and Moyer (2009)” in Line 175.
Response 5: The missing references are now included in the revised paper as follows;
- Pandey (2001) is found on the reference list in LINE 579-580
- Adrangi, Chatrath and Sanvicente (2000) is found on the reference list in LINE 466-467
- McGuigan, Kretlow and Moyer (2009) is found on the reference list in LINE 568